# Spray-Drying Microencapsulation of *Bauhinia ungulata* L. var. *obtusifolia* Aqueous Extract Containing Phenolic Compounds: A Comparative Study Using Different Wall Materials

**DOI:** 10.3390/pharmaceutics16040488

**Published:** 2024-04-02

**Authors:** Myrth Soares do Nascimento Remígio, Teresa Greco, José Otávio Carréra Silva Júnior, Attilio Converti, Roseane Maria Ribeiro-Costa, Alessandra Rossi, Wagner Luiz Ramos Barbosa

**Affiliations:** 1Laboratory of Chromatography and Mass Spectrometry, Graduate Program in Pharmaceutical Innovation, Institute of Health Sciences, Federal University of Pará, Belém 66075-110, Brazil; myrthsoares@ufpa.br; 2Department of Food and Drug, University of Parma, 43124 Parma, Italy; 3Laboratory of R&D Pharmaceutical and Cosmetic, Graduate Program in Pharmaceutical Innovation, Institute of Health Sciences, Federal University of Pará, Belém 66075-110, Brazil; carrera@ufpa.br; 4Department of Civil, Chemical and Environmental Engineering, Pole of Chemical Engineering, University of Genoa, 16145 Genoa, Italy; converti@unige.it; 5Laboratory of Nanotechnology, Graduate Program in Pharmaceutical Innovation, Institute of Health Sciences, Federal University of Pará, Belém 66075-110, Brazil; rmrc@ufpa.br

**Keywords:** phytotherapy, medicinal plants, radical scavenger, phenolic compounds, medicinal tea, microencapsulation

## Abstract

Species belonging to the *Bauhinia* genus, usually known as “pata-de-vaca”, are popularly used to treat diabetes. *Bauhinia ungulata* var. *obtusifolia* (Ducke) Vaz is among them, of which the leaves are used as a tea for medicinal purposes in the Amazon region. A microencapsulation study of lyophilized aqueous extract from *Bauhinia ungulata* leaves, which contain phenolic compounds, using five different wall materials (maltodextrin DE 4-7, maltodextrin DE 11-14; β-cyclodextrin; pectin and sodium carboxymethylcellulose) is described in this paper. The microstructure, particle size distribution, thermal behavior, yield, and encapsulation efficiency were investigated and compared using different techniques. Using high-performance liquid chromatography, phenolics, and flavonoids were detected and quantified in the microparticles. The microparticles obtained with a yield and phenolics encapsulation efficiency ranging within 60–83% and 35–57%, respectively, showed a particle size distribution between 1.15 and 5.54 µm, spherical morphology, and a wrinkled surface. Among them, those prepared with sodium carboxymethylcellulose or pectin proved to be the most thermally stable. They had the highest flavonoid content (23.07 and 21.73 mg RUTE/g Extract) and total antioxidant activity by both the DPPH (376.55 and 367.86 µM TEq/g Extract) and ABTS (1085.72 and 1062.32 µM TEq/g Extract) assays. The chromatographic analyses allowed for quantification of the following substances retained by the microparticles, chlorogenic acid (1.74–1.98 mg/g Extract), p-coumaric acid (0.06–0.08 mg/g Extract), rutin (11.2–12.9 mg/g Extract), and isoquercitrin (0.49–0.53 mg/g Extract), compounds which considered to responsible for the antidiabetic property attributed to the species.

## 1. Introduction

*Bauhinia ungulata* L. var. *obtusifolia* (Ducke) Vaz, vernacular “pata-de-vaca,” is a tree from the Leguminosae family, considered to be endemic to Brazil, of which occurrence has been confirmed in the Brazilian North Amazonia and Northeast regions [1,2,3]. The leaves, like those of other species of the genus, are empirically used as an infusion to treat diabetes [4,5]. The chemical constitution and biological activities of this infusion have scarcely been investigated, with a single report describing its antioxidant activity which was detected by using the DPPH^·^ and β-carotene/linoleic acid radical methods as a probable result of the presence of phenolic compounds [6].

Phenolic compounds are considered to be the main group of substances synthesized by the secondary metabolism of plants [7]. Due to their chemopreventive/protective roles, consumption of these compounds reduces the incidence of many chronic diseases induced by oxidative stress, such as neurodegenerative and cardiovascular diseases, cancer, inflammation, infections, and diabetes. Products rich in polyphenols can modulate carbohydrate and lipid metabolism, attenuate hyperglycemia and dyslipidemia, improve the function of pancreatic β-cells, stimulate insulin secretion, and reduce resistance to this hormone [8].

However, phenolic compounds are sensitive to adverse environmental conditions, including light, temperature, pH, moisture, and oxygen, thus making them susceptible to degradation during processing and storage [9]. In this sense, it is important to protect them to preserve their biological activities and properties. Improving their bioaccessibility and bioavailability, as well as promoting their transport for absorption by the human body, are also of great interest [10].

For this purpose, various microencapsulation strategies have been introduced, among which spray-drying is one of the commonly most used since it ensures high-quality products in addition to being a relatively low-cost process [11,12]. This technique has proven to be an effective method for drying and encapsulating phenolic compound-rich materials, such as extracts from *Litsea glaucescens, Camellia sinensis, Prunus salicina, Crocus sativus,* and *Laurus nobilis*, among others [13,14,15,16,17].

Overall, the addition of appropriate wall material to the solution to be sprayed should provide high solubility, low hygroscopicity, effective emulsification, flavor masking capability, good film formation, and reduced process costs. Its selection must also consider the specific characteristics of the natural bioactive substances to be microencapsulated, such as sensory properties, physicochemical stability, bioactive retention, loading, and release [12]. Wall materials used to encapsulate phenolic compounds include polysaccharides, such as maltodextrin with different dextrose equivalent (DE) values [13,18,19,20,21,22], β-cyclodextrin [11,15,23,24,25], pectin [26,27], and cellulose or its derivatives like sodium carboxymethylcellulose [28,29]. However, when it comes to preparing pharmaceutical formulations, such as tablets, from the encapsulated extracts, the percentage of excipients to be added becomes a critical manufacturing issue, as tablets often contain a high dose of dry plant extract [30].

To the best of our knowledge, there is no available information on an eligible wall material for the encapsulation of aqueous extract containing phenolic compounds from *B. ungulata*. Therefore, since the coating behavior of each wall material is different, their suitability for encapsulation needs to be experimentally assessed.

So, this work reports the preparation of microparticles loaded with the dried aqueous extract of *B. ungulata* var. *obtusifolia* leaves by spray-drying using different wall materials, aiming to stabilize the extract and its active compounds and to improve their bioavailability if orally administrated. The morphology, particle size distribution, thermal behavior, process yield, phenolic content, flavonoid content, and encapsulation efficiency of microparticles were investigated.

## 2. Materials and Methods

### 2.1. Materials

The extract used in this study was obtained from a plant material collected in November 2017 at Castanhal, Pará state, Brazil (coordinates −1.297160, −47.921180), botanically described by Manoel dos Reis Cordeiro (Empresa Brasileira de Pesquisa Agropecuária—EMBRAPA, Belém, Brazil), one herborized specimen of which was deposited at the Herbarium of Embrapa Amazônia Oriental (IAN) under the number 196.015.

The standards rutin, chlorogenic acid, p-coumaric acid, isoquercitrin, gallic acid, 2,2-diphenyl-1-picrylhydrazyl (DPPH), 2,2-azinobis (3-ethylbenzothiazoline-6) sulfonic acid (ABTS), 6-hydroxy-2,5,7,8-tetramethylchroman-2-carboxylic acid (Trolox), maltodextrin DE 4-7, and Folin–Ciocâlteu reagent were obtained from Sigma Aldrich (St. Louis, MI, USA). Potassium persulphate and aluminum chloride were purchased from LabSynth (São Paulo, Brazil), while maltodextrin Lycatab DSH DE 11-14 and β-cyclodextrin were obtained from Roquette (Lestrem, France). Pectin LM-22-CG and sodium carbonate were purchased from CP Kelco GENU (Großenbrode, Germany) and A.C.E.F. (Fiorenzuola D’Arda, Italy), respectively, and sodium carboxymethylcellulose was donated by Lisapharma (Erba, Italy). Acetic acid was obtained from VWR Chemicals (Fontenay-sous-Bois, France), while formic acid was purchased from Vetec (Duque de Caxias, Brazil). All analytical-grade solvents were purchased from VWR Chemicals and Merck (Darmstadt, Germany).

### 2.2. Methods

#### 2.2.1. Preparation of Lyophilized Aqueous Extract

Ground leaves were extracted by infusion in boiling distilled water (50 g/L) for 30 min [31]. The mixture was filtered under reduced pressure and then concentrated to about 200 mL using a rotary evaporator (model R-210, Buchi, Flawil, Switzerland) under low pressure at 40 °C. The concentrated extract was frozen at −40 °C and subsequently freeze-dried in a Liotop L101 instrument (Liobras, São Carlos, Brazil) at −50 °C and at a pressure between 50–500 µHg to obtain the lyophilized extract (Bu-L). The residual water content in Bu-L was gravimetrically determined in triplicate, according to the method reported in the Brazilian Pharmacopoeia [32], and the results were expressed as % *w*/*w*.

#### 2.2.2. Preparation of the Encapsulated Extract

Solutions containing 51 mg of each of the encapsulating agents, namely maltodextrin DE 4-7 (MD4), maltodextrin DE 11-14 (MD11), β-cyclodextrin (βCD), pectin (Pec), and sodium carboxymethylcellulose (CMC), were prepared in a flask with 56 mL of purified water. Then, 500 mg of Bu-L was added to each solution under stirring with a magnetic bar until complete dissolution. A solution without the addition of an encapsulating agent was prepared as a control.

##### Determination of the Solution’s Viscosity

The viscosities of the solutions to be sprayed were measured in triplicate at 20 °C using a viscometer (Rotational Smart R, Fungi-Lab, Barcelona, Spain) equipped with a spindle LCP at 100 rpm.

##### Spray-Drying of the Solutions

Solutions were spray-dried in duplicate in a MiniSpray-Dryer (B-290, Büchi, Milan, Italy) that was equipped with an inert loop, under a nitrogen atmosphere, using a two-fluid nozzle with a 1.4 mm diameter orifice. The temperature of the inlet drying air was 150 °C, the feed volumetric flow rate was 4 mL/min, the drying air volumetric flow rate was 742 L/h, and the aspiration rate was 90% [33], while the outlet air temperature ranged from 60 to 75 °C. The spray-dried samples obtained were labeled as Bu-MD4, Bu-MD11, Bu-βCD, Bu-Pec, Bu-CMC, and Bu-A (without any wall material) and stored in sealed glass vials, with a rubber stopper and aluminum cap, in a desiccator at room temperature (25 ± 1 °C) to avoid moisture absorption.

#### 2.2.3. Yield of Spray-Drying Process

The spray-dried (SD) microparticles were quantitatively recovered from the product collection vessel and cyclone and weighed on an analytical balance (model Crystal 200 CAL CE, Gibertini, Novate Mil.se (MI), Italy). The yield of the spray-drying process (YD) was determined as a percentage, considering the weight of total solids in the sprayed solution and that of the SD microparticles obtained after each spray-drying cycle, using the following equation:(1)$$YD=\frac{weight\;of\;SD\;microparticles\;(mg)}{weight\;of\;total\;solids\;(mg)}\times 100$$

#### 2.2.4. Characterization of the SD Microparticles

##### Morphology

The morphological and surface characteristics of Bu-A, Bu-MD4, Bu-MD11, Bu-βCD, Bu-Pec, and Bu-CMC microparticles were examined by using scanning electron microscopy (SEM, Zeiss AURIGA, Oberkochen, Germany) at an extra-high tension of 1.00 kV. The samples were placed on a double-sided adhesive tape pre-mounted on an aluminum stub and analyzed after 30 min depressurization.

##### Particle Size Distribution

About 5 mg of each sample of SD microparticle was suspended in 10 mL of solution 1% (*w*/*v*) of sorbitan trioleate (Span 85) in cyclohexane and then dispersed in an ultrasonic bath (model 8510, Branson Ultrasonics Corporation, Brookfield, CT, USA) at high potency for 2 min to avoid the microparticles aggregating. The particle size distribution was measured, in triplicate, at an obscuration threshold of at least 10%, using a laser light diffractometer (model Spraytec, Malvern Instruments Ltd., Malvern, UK). The results were expressed in terms of median volume diameter (D_v50_), 10th (D_v10_), and 90th (D_v90_) percentiles, i.e., the values of each diameter to which 50, 10, and 90% of the population are below, respectively. The span value was also calculated as (Dv_90_ − Dv_10_)/Dv_50_.

#### 2.2.5. Assays for the Quantitative Determination of Phenolic Compounds

##### Total Phenolics Retained in the SD Microparticles

The samples for the analysis were prepared following the method reported by Robert et al. [34], with modifications. Briefly, about 20 mg of accurately weighed Bu-MD4, Bu-MD11, and Bu-βCD were suspended in 2 mL of 50:8:42 (*v*/*v*/*v*) methanol/acetic acid/ultrapure water solution, stirred for 1 min (Vortex Velp Scientifica, Milan, Italy), sonicated for 40 min (Branson 8510 Ultrasonic Cleaner, Sterling Heights, MI, USA), and then centrifuged at 18,670× *g* for 30 min (Scilogex D3024 Centrifuge, Giorgio Bormac S.r.l., Carpi, Italy). The resulting supernatants were filtered on a 0.45 µm hydrophilic membrane. The same procedure was applied to Bu-Pec, Bu-CMC, and Bu-A but using a 50:50 (*v*/*v*) methanol/ultrapure water solution as solvent. Membranes with a 0.45 µm pore diameter were used for filtration, except for Bu-Pec, for which a 0.22 µm membrane was used. The Total Phenolics Retained (TPR) in the SD microparticles were quantified according to the method described by Dewanto et al. [35], described in the Section “Determination of Total Phenolic Content in the Microparticles”. The total phenolic content in Bu-L (TCE) was used as a Ref. [36].

##### Total Phenolic Compounds on the Microparticle Surface

The samples for the analysis were prepared following the method reported by Robert et al. [34], with modifications. Briefly, about 20 mg, accurately weighed, of each type of microparticle was treated with 2 mL of ethanol/methanol (50:50 *v*/*v*), stirred for 1 min in Vortex, and filtered on a 0.45 µm hydrophilic membrane, except for Bu-Pec, where a 0.22 µm hydrophilic membrane was used. The total phenolic compounds on the surface of the microparticles (PS) were quantified according to the method described by Dewanto et al. [35], as described in the Section “Determination of Total Phenolic Content in the Microparticles”. The total phenolic content in the lyophilized extract (TCE) was used as a reference [36].

##### Determination of Total Phenolic Content in the Microparticles

Aliquots of 125 µL of each solution, prepared for the determination of TPR or PS, and 125 µL of Folin–Ciocalteu reagent were added to 0.5 mL of ultrapure water and left to stand at room temperature for 6 min; then, 1.25 mL of 7% sodium carbonate aqueous solution and 1 mL of ultrapure water were added. The samples were kept at room temperature for 90 min and were then analyzed by utilizing UV-Vis spectrophotometry (Lambda 25 UV/Vis Perkin Spectrophotometer, Monza, Italy) at 760 nm, according to Dewanto et al. [35], by using gallic acid as a standard (50–250 µg/mL, y = 0.0346x − 0.0542).

For the TPR determination, 20 parts of Bu-A, Bu-MD4, Bu-MD11, Bu-βCD, Bu-CMC, and Bu-Pec were diluted with 80 parts (*v*/*v*) of methanol/water (50:50 *v*/*v*), as well as for samples which were used for PS determination, which were diluted at 1:1 (*v*/*v*), to set their absorbance values into the calibration curve range of gallic acid. All analyses were performed in triplicate, and the values were expressed in milligrams of gallic acid equivalent per gram of extract (mg GAEq/g Extract).

##### Determination of Total Flavonoid Content

The determination of the flavonoid content was performed using the modified method reported by Costa et al. [37]. About 20 mg of accurately weighed Bu-MD4, Bu-MD11, and Bu-βCD, suspended in 2 mL of methanol/acetic acid/water (50:8:42 *v*/*v*/*v*), and of Bu-Pec, Bu-CMC, and Bu-A, suspended in 2 mL of methanol/water (50:50 *v*/*v*), were used for quantifying the total flavonoids in the SD microparticles. The resulting dispersions were stirred for 1 min using a vortex, sonicated for 40 min (Cristófoli Cleaner, Campo Mourão, PR, Brazil), and centrifuged at 7043× *g* for 15 min (Eppendorf Centrifuge 5804, Hamburg, Germany). The supernatants were filtered on a 0.22 µm hydrophilic membrane; then, once at a volume of 400 µL, 500 µL of 2.5% aluminum chloride solution and 4.1 mL of distilled water were added to each sample. The obtained solutions were stored in the dark for 30 min before being analyzed at 425 nm using a Shimadzu UV-1800 spectrophotometer (Kyoto, Japan). The total flavonoid content was calculated based on a standard curve of rutin (5–30 µg/mL, y = 0.0051x + 0.0805) and expressed in milligrams of rutin equivalent for gram of extract (mg RUTE/g Extract). The reference value was determined for the extract [36].

#### 2.2.6. Total Antioxidant Activity

The solutions, used to quantify the flavonoid content, were further evaluated for their total antioxidant activity against the free radicals DPPH^∙^ and ABTS^∙+^ after a dilution to 1 mg/mL with methanol/water (50:50 *v*/*v*). The diluted solutions were analyzed with a spectrophotometer (Shimadzu UV 1800, Kyoto, Japan) and the results are expressed as a percentage of inhibition (Equation (2)) and total antioxidant activity (TAA) (Equation (3)).
(2)$$\%\;Inhibition=\frac{\left(Abs\;control-Abs\;sample\right)}{Abs\;control}\times 100$$
(3)$$TAA=\frac{1000\times (Abs\;Trolox\;1000\;µM)}{sample\;concentration\times Abs\;sample}$$
where Abs is the absorbance value, and Abs Trolox is the absorbance of Trolox at 1000 µM, estimated from the standard curves; the concentration of the sample is expressed in g/L. The TAA equation is based on Rufino et al. [38].

##### DPPH Assay

Aliquots of 150 µL of each diluted sample, as described in the Section “Determination of Total Flavonoid Content”, were added to 5850 µL of DPPH^∙^ radical (2,2-diphenyl-1-picrylhydrazyl at 0.06 mM in methanol). The solutions were incubated in the absence of light for 30 min and then analyzed in triplicate at 515 nm with the spectrophotometer [39]. The antioxidant activity was calculated using a Trolox standard curve (50–1000 µM, y = −0.0005x + 0.6516) and the results were expressed as µM Trolox equivalent (µM TEq/g Extract) and as a percentage of inhibition, using a reference value determined for the extract [36].

##### ABTS Assay

Aliquots of 30 µL of each diluted sample were added to 3000 µL of an ABTS^∙+^ radical solution (obtained from 88 µL of 1.40 mM potassium persulfate solution and 5 mL of 7 mM ABTS solution, kept in the dark at room temperature for 16 h), incubated in the absence of light for 6 min, and then analyzed in triplicate at 734 nm with the spectrophotometer [38]. The antioxidant activity was calculated using a Trolox standard curve (100–2000 µM, y = −0.0003x + 0.7131). The results are expressed as µM Trolox equivalent (µM TEq/g Extract) and as a percentage of inhibition, using a reference value determined for the extract [36].

#### 2.2.7. Encapsulation Efficiency

The encapsulation efficiency (EE) corresponds to the percentage of phenolics within each SD microparticle sample and was calculated for Bu-MD4, Bu-MD11, Bu-βCD, Bu-Pec, and Bu-CMC from the amounts of phenolic compounds retained by the microparticles (TPR) minus those present on their surface (PS) relating to the total phenolic content in the lyophilized extract (TCE) [40] by using the following equation:(4)$$EE=\frac{TPR-PS}{TCE}\times 100$$

#### 2.2.8. Encapsulation Yield of the Phenolic Compounds

The yield of the encapsulation process (YE) is defined as the ratio of the percentage of phenolics, encapsulated by each wall material (TPR), to the total phenolic compounds in the lyophilized extract (TCE) [41,42], according to Equation (5):(5)$$YE=\frac{TPR}{TCE}\times 100$$

#### 2.2.9. Thermogravimetric Analysis

Accurately weighed samples of 3 to 6 mg of each SD microparticle were placed in a 70 µL aluminum pan, covered with a perforated lid, and heated from 25 to 200 °C or 250 °C (according to their decomposition temperatures) at a scanning rate of 20 °C/min under N_2_ atmosphere (80 mL/min) using a TGA/DSC1 instrument (Mettler Toledo, Milan, Italy), driven by STARe Software. This software allows for the comparison of the thermogravimetric curve (TGA), used to measure the weight loss in the sample during heating, with the derivative thermogravimetric curve (DTG), showing the rate of change in mass concerning temperature or time.

#### 2.2.10. Differential Scanning Calorimetry

Samples of 3 to 6 mg, accurately weighed, of each SD microparticle were placed in a 40 µL aluminum pan with a pin, sealed with a lid that was pierced twice, to evaluate their thermal behavior. Differential scanning calorimetry (DSC) measurements were performed using a DSC 821e instrument (Mettler Toledo, Schweiz, Switzerland) driven by the STARe software. The scans were performed between 25 and 200 °C or 250 °C (depending on sample decomposition temperature) at a rate of 20 °C/min under a N_2_ atmosphere (100 mL/min).

#### 2.2.11. Qualitative and Quantitative Analyses of Compounds in the Microparticles Using High-Performance Liquid Chromatography

##### Chromatographic Conditions

Detection and quantification of the phenolic compounds were performed at 40 °C in an Agilent 1260 Infinity (Wilmington, DE, USA) instrument through high-performance liquid chromatography with a diode array detector (HPLC-DAD) using a Zorbax SB-Aq C18 column (150 × 4.6 mm, 5 µm). The mobile phase consisted of (A) 0.1% (*v*/*v*) aqueous formic acid (pH = 3) and (B) acetonitrile. The elution was in gradient mode: 0–60 min, 1–20% B; 60–65 min, 20–1% B. The HPLC analysis was conducted at a flow rate of 1 mL/min, injection volume of 20 µL, and detection wavelength of 330 nm [36].

##### Qualitative Analyses of the Compounds

The presence of chlorogenic acid, p-coumaric acid, isoquercitrin, and rutin in the SD microparticles was investigated using 0.1 mg/mL solutions of the samples by comparison of their UV spectra with those relative to different peaks in the extract at the corresponding retention times. The compounds were detected and co-eluted with the extract in determined amounts, and the increased peak areas corresponding to each one were registered and compared to the peak areas in the non-enriched extract [36].

##### Quantitative Analyses of the Compounds

Stock solutions in methanol/water (50:50 *v*/*v*) for chlorogenic acid (1 mg/mL) and p-coumaric acid (1 mg/mL), and in methanol for rutin (1 mg/mL) and isoquercitrin (0.2 mg/mL), were used to prepare the working solutions containing 10, 30, 100, and 400 µg/mL of p-coumaric acid, isoquercitrin, chlorogenic acid, and rutin, respectively. The working solutions were further diluted in methanol according to the concentrations listed in Table 1. The calibration curves were plotted using the average peak areas, collected in triplicate, and the following metrics from the calibration curves the regression equation, linearity, coefficient of determination (R^2^), limit of detection (LOD), and limit of quantification (LOQ) were determined: chlorogenic acid (y = 61.122x + 0.1721, 2.5–50 µg/mL, R^2^ = 0.9999, LOD 0.1878 µg/mL, and LOQ 0.5691 µg/mL), p-coumaric acid (y = 82.789x + 4.0462, 0.25–5.00 µg/mL, R^2^ = 0.9995, LOD 0.0614 µg/mL, and LOQ 0.1861 µg/mL), rutin (y = 23.827x + 12.541, 10–200 µg/mL, R^2^ = 0.9998, LOD 1.6892 µg/mL, and LOQ 5.1189 µg/mL), isoquercitrin (y = 43.851x + 7.1778, 0.75–15 µg/mL, R^2^ = 0.9997, LOD 0.1960 µg/mL, and LOQ 0.5938 µg/mL). The equations were then used to determine the concentration of these compounds in the microparticles. The reference values were determined for Bu-L [36].

#### 2.2.12. Statistical Analysis

Statistical analysis was performed using GraphPad Prism 9.00 (Boston, MA, USA, 2020). One-way analysis of variance (ANOVA) and Tukey’s multiple comparisons test were conducted to detect statistically significant differences (*p* < 0.05) among values. The results of the statistical analysis are represented by lowercase letters to the right of each mean value (a, b, c, d, e, or their combinations). Equal letters in the same column indicate no statistical difference between the values compared. *p* values < 0.05 are available in the Appendix A.

## 3. Results

### 3.1. Preparation of Spray-Drying Microparticles

The solutions used for the spray-drying process contained 51 mg of wall material, showed a solid content of approximately 0.9% (Table 2), and the viscosity for all samples was quite similar, except for Bu-CMC, for which the value was higher (*p* < 0.0001).

### 3.2. Yield of Spray-Drying Process

All spray-dried microparticles had a yield greater than 60% (Table 2). The powders without wall material (around 64%) and Bu-MD4 (about 89%) showed the lowest and highest yield, respectively. On the other hand, for the Bu-MD11, Bu-βCD, Bu-Pec, and Bu-CMC microparticles, the YD ranged from 72 to 83%.

### 3.3. Morphology and Particle Size Distribution

Bu-A, Bu-MD4, Bu-MD11, Bu-βCD, Bu-Pec, and Bu-CMC were analyzed by using SEM (Figure 1), and it was observed that their particles were somewhat agglomerated and had a round and non-porous surface. The SD microparticles are heterogeneous, predominantly wrinkled structures with many shrinkages in all samples, except for Bu-Pec, where the morphology of the microparticles seems smoother.

All the microparticles showed a spherical shape of variable sizes, ranging from 1.15 to 5.54 µm, with a volumetric average particle diameter (D_v50_) in the range of 2.01–2.87 µm, with the Bu-βCD and Bu-CMC particles showing the smallest and largest particle diameters, respectively (*p* < 0.05) (Table 3).

All the samples tended toward polydispersion, as seen by the peaks corresponding to the particle diameter of the analyzed microparticles (Figure 2). The single curves are reported in the Appendix A. Each peak represented a predominant particle size, except for Bu-MD4, which showed only one peak. Regarding the particle size distribution range represented by the span values, similar values were found among these values (*p* > 0.05) according to the analysis of variance performed.

### 3.4. Thermal Behavior of the Microparticles

The TGA/DTG and DSC techniques allowed us to evaluate the thermal behavior of the microparticles. Two representative stages were observed in the thermogravimetric curves (Figure 3). The first significant weight loss occurred between 50 and 120 °C and can be attributed to the residual moisture content of the SD samples, which ranged between 2.90 and 4.25% (Table 4). However, the values were lower than that found for Bu-L (7.9 ± 0.5%). The use of wall materials promoted a significant increase in the residual water content of the microparticles compared to Bu-A, but only for Bu-MD11, Bu-Pec, and Bu-CMC this difference was statistically significant (*p* < 0.05). After the initial water loss event from the samples, a pronounced mass loss was observed in the TGA/DTG curves at 150 °C (Bu-A, EBu-MD4, Bu-MD11, and Bu-βCD) or at 190 °C (Bu-CMC and Bu-Pec), corresponding to the sample decomposition.

The DSC thermograms of the SD microparticles, as shown in Figure 4 and the Appendix A, exhibited two main thermal events. The first broad event, which generally occurred between 50 and 120 °C, is attributed to the moisture evaporation in agreement with the behavior observed in the thermogravimetric curves. A variation in the baseline in the first part of the endothermic event can be attributed to the glass transition (T_g_) of the microparticles, determined by their midpoint temperature [43]. Although the extracts have different T_g_ values, statistical analysis showed that there was no significant difference between the T_g_ values recorded (*p* > 0.05).

The second event recorded in the thermograms occurred in the exothermic direction and is associated with the decomposition of the SD samples. For Bu-A, Bu-MD4, Bu-MD11, and Bu-βCD (Figure 4A), this event started at temperatures of around 160 °C (Table 4) which were not statistically different from each other (*p* > 0.05). In the curves of the microparticles prepared with Pec or CMC (Figure 4B), this event started at 191 and 188 °C, respectively. The latter values are comparable to each other (*p* > 0.05) and are significantly higher than those recorded for the other samples (*p* < 0.0001).

### 3.5. Total Phenolic Content, Total Flavonoid Content, Yield of Encapsulation, and Encapsulation Efficiency

The samples were analyzed for the content of phenolic and flavonoid compounds present by determining TPR, PS, and TFC. For comparison purposes, all values were adjusted according to the residual moisture content determined for each SD microparticle sample. The TPR and TFC in the microparticles ranged from 69.98 to 101.28 mg GAEq/g Extract and 12.69 to 23.07 mg RUTE/g Extract, respectively. Therefore, the determination of the total phenolic content in the microparticle and the total phenolic content on the surface of the microparticle allowed us to calculate the encapsulation efficiency of the encapsulated extract. Among them, Bu-MD11, Bu-CMC, and Bu-βCD showed a higher encapsulation efficiency of phenolics. In terms of yield of encapsulation, the microparticles obtained with MD4, MD11, and βCD exhibited better results (83.09, 89.71, and 83.59%, respectively), with values statistically close to each other (*p* > 0.05) (Table 5).

### 3.6. Antioxidant Activity

The samples were able to scavenge the radicals tested, showing different percentages of inhibition and total antioxidant activity (TAA) depending on the method used or the sample considered (Table 6). In the evaluation of DPPH^∙^ scavenging, Bu-L was the sample with the lowest total antioxidant activity compared to the others, with only 308.67 ± 2.64 µm TEq/g Extract. Still, the statistical analysis of the data shows that this value was not significantly different from Bu-A (*p* > 0.05). The data also show that all SD microparticles in which the extract was encapsulated with a wall excipient had a higher TAA than that calculated for the unencapsulated extract. In terms of ABTS^∙+^ radical scavenging, Bu-CMC and Bu-Pec were the most active.

### 3.7. Characterization and Quantification of Phenolic Compounds in the Microparticles HPLC-DAD

The analysis of SD microparticles by using HPLC-DAD showed chromatographic peaks corresponding to those observed in Bu-L [36]. The average contents of p-coumaric acid and isoquercitrin determined in the extract were also observed in the microparticles, regardless of the spray-drying process or the wall agent used, showing no statistically significant difference between the values (*p* > 0.05). In contrast, the content of chlorogenic acid and rutin was variable (Table 7). The spray-dried extract without wall material, Bu-A, showed 1.975 ± 0.052 and 12.886 ± 0.115 mg/g Extract of phenolic acid and flavonoid, respectively; these values are significantly higher than those determined for all the other spray-dried samples (*p* < 0.05). On the other hand, among the microparticles, Bu-Pec and Bu-CMC had the highest content of chlorogenic acid and rutin, significantly differing from the values obtained for all the other spray-dried samples (*p* < 0.05), except for Bu-βCD, which showed comparable levels of these compounds.

## 4. Discussion

### 4.1. Spray-Drying Process

The viscosity of the solutions to be spray-dried influences the drop formation and consequently modifies the particle size at the end of the process. Furthermore, less-viscous solutions demand lower pressure to form the spray, saving energy [44]. Among the solutions prepared for atomization, only Bu-CMC differed from the others in terms of viscosity, inferring that the solid content did not significantly influence the viscosity of the solutions. The viscosity value determined for Bu-A, with a lower solid content, was similar to those measured for Bu-MD4, Bu-MD11, Bu-βCD, and Bu-Pec, which infers that viscosity is strongly influenced by the type of encapsulating agent used, as the solution with CMC showed an increased viscosity due to the thickening property of this excipient that, like other cellulose derivatives, acts as a modifier of viscosity [45]. Although pectin has thickening and gelling properties too, its addition to solutions did not lead to any significant change in viscosity, likely since the low methoxylated pectin used in this work would have required calcium ions to display such functions [46]. Despite the difference observed in this property between the solution prepared with CMC and the others, all had suitable viscosity for atomization since they flowed easily through the equipment during drying.

The applied method can be considered efficient since the YD values are higher than 50%, a value that defines the efficiency of a bench spray-drying process [21,47]. The lowest yield of Bu-A suggests that using wall materials favors a higher solid recovery by this drying process, especially when using MD4, for which the yield was higher than 88%. This was an expected result, since the addition of these agents to solutions of plant extracts to be sprayed decreases the stickiness of the sample, favoring less deposition of the atomized material on the walls of the drying chamber and increasing the process yield [48,49]. The stickiness of solutions is related to the high content of low-molecular-weight sugars and organic acids, and this reduces their glass transition temperature (T_g_), promoting cohesion between particles or their adhesion to dry surfaces. The addition of excipients, such as maltodextrins, induces a material T_g_ increase, thus facilitating its transit through the cyclone chamber, consequently increasing the process yield [19,50]. A similar result was described by Vidovic et al. when drying a liquid *Satureja montana* extract, as they observed a significant amount of material deposition in the drying chamber during the atomization [21]. The addition of maltodextrin DE-16 at 10%, 30%, and 50% to the extract improved the efficiency of the process and promoted similar yields to each other, ranging from 66 to 68%. Cid-Ortega and Guerrero-Beltrán have also reported higher yields in the microencapsulation of *Hibiscus sabdariffa* (Roselle) extract after adding maltodextrin [51]. The authors recorded 58.19% solid recovery without the encapsulating agent and reached 56.46, 59.81, and 72.07% by the addition of 3, 5, and 10% of the wall material, respectively.

Vladic et al. spray-dried a liquid extract from *Achillea millefolium* with and without maltodextrin DE16.5-19.5 [22], which performed similar drying efficiencies (around 70%), contrasting with the results observed by other authors. A lower recovery was registered by the atomization of the extract concentrated before drying, a step which, according to the authors, could reduce the efficiency of the process due to the higher solid content. Increasing the solid content in a solution alters its viscosity and may decrease the microparticle yield, as Tonon et al. reported by obtaining *Euterpe oleraceae* Mart. microparticles through spray-drying using maltodextrin DE9-12 as the wall material [48].

The yield values of Bu-MD4 and Bu-MD11, which were higher than 70%, agree with those observed by Caliskan and Dirim [19]. They spray-dried *Rhus coriaria* L. extract with maltodextrin DE 10-12, yielding 70.21%, 86.77%, 97.45%, and 98.5% for solutions with soluble solid contents of 10, 15, 20, and 25%, respectively, indicating that high amounts of wall material increase the efficiency of spray drying. Similar yields were obtained by Farias-Cervantes et al. when spraying raspberry or blackberry juices with maltodextrin, as they revealed YD values ranging from 52 to 66% and 64 to 78%, respectively [52]. Nevertheless, the reported YD values were higher than those observed by Costa and collaborators for the microencapsulation of a cupuassu seed by-product extract with maltodextrin that ranged from 11 to 19% with 5, 7.5, and 10% (*w*/*v*) of wall material [37]. In addition to the different chemical compositions of the extracts, the distinct results could be due to the operating process conditions of the spray-dryer, the degree of dextrinization, the proportion of maltodextrin added to the solutions, and the core/wall ratio of the microparticles obtained.

The yield recorded in this work for Bu-MD4, which was the highest observed, could have been due to the low dextrinization grade of the maltodextrin used, which is mainly composed of long polysaccharides of high molecular weight and a small fraction of oligomers. The maltodextrin DE 4-7 can form less sticky solutions, favoring the yield of microparticles prepared by spray drying, as observed by Siemons et al. [53].

The similar spray-drying yields of Bu-βCD, Bu-Pec, and Bu-CMC were higher than those described in other works using the same wall materials. In the case of Bu-CMC, the yield was higher than those observed for the encapsulation of *Lannea microcarpa* (46.7%) and *Morus alba* (53.47–61.87%) extracts [29,54]. Studies in which β-CD and Pec were used to obtain atomized extracts of *Sideritis stricta* and *Gentiana asclepiadea*, reported yields of 33.07% and 49.51–62.12%, respectively, indicating lower spray-drying efficiencies in comparison to the present work [55]. These variations could be due to the different compositions of the feed solutions and the selected process parameters.

### 4.2. Morphology and Particle Size of Spray-Dried Microparticles

Regarding the characterization of the microparticle structure, morphology is a relevant parameter as it reveals the protective capacity of different polymers and reflects the quality of the microparticles obtained. The desirable morphological characteristics of the particles produced by spray-drying, which affect the stability of the encapsulated phytochemicals compounds, include a smooth surface without structural defects, compounds in the core of the particles, and a size as large as possible [56]. The particle size depends on the intended use of the microparticles obtained by spraying; for example, microparticles for direct compression tend to have a size distribution in the range of 100–200 µm, while for inhalation this size range should be between 1 and 5 µm [57].

The agglomeration of some microparticles in the Bu-A, Bu-MD4, Bu-MD11, Bu-βCD, Bu-Pec, and Bu-CMC samples, as shown in the SEM images (Figure 1), is a phenomenon that can occur when the T_g_ of a material is lowered by the influence of its plasticization with water. As a result, the material becomes sticky on the surface and promotes particle cohesion during the spraying process [58].

The spherical shape of the analyzed microparticles is a characteristic attribute of spray-dried products [11,59]. The integrity of the particles in the samples and the absence of pores on their surface can be considered a technical advantage, as they increase the retention and protection of the active compounds inside the particles. The occurrence of pores or cracks on the surface can expose the particle core to the atmosphere and allow leaching or migration of the actives to the surface where, in contact with oxygen, they can undergo degradation [11,18]. Regarding the surface morphology, the wrinkling of the microparticle may be due to the drastic moisture loss following the rapid evaporation of the atomized droplets in contact with the hot air or inert gas during the spray-drying process [60]. Similar aspects of microparticles prepared with maltodextrin, cyclodextrin, or a combination of these encapsulating agents have been observed [23,24].

Sansone et al. have described the obtaining of spherical particles with a smooth surface using 50% (*w*/*w*) of CMC for the *Lannea microcarpa* encapsulation by spray-drying [54]. The Bu-CMC microparticle surface differs from those previously cited, probably because the amount of wall material used to encapsulate *B*. *ungulata* extract represents only 10% of the total solid content. In addition, both the spray-drying process parameters and the composition of the extracts play decisive roles in the microparticle surface.

The larger particles observed in the Bu-CMC microparticles were foreseen because of the higher viscosity of the feed solutions prepared with this wall excipient when compared to other microparticles, since the higher the viscosity of the liquid, the larger the droplets formed during atomization and, therefore, the larger the particles obtained after drying [48]. This result is also related to the one published by Jinapong et al. for instant soy milk powder produced by ultrafiltration and spray drying [61]. Keogh et al. also observed a linear increase in the particle size as the viscosity of the feed solution, an ultrafiltered concentrate integral milk, increased [62]. In both studies, the authors attributed the large particle size to the high viscosity of the feed solutions.

Based on the particle size distribution data, the tendency of a monomodal behavior for Bu-MD4 can be inferred, while a multimodal distribution is observed for the other microparticle samples. It is worth noting that smaller particles can penetrate the spaces among the larger ones, occupying less space and increasing the density of the microparticles. The presence of “populations” of larger particles can be attributed to the particles’ agglomeration. Regarding the particle size distribution range, the data indicate a narrow distribution in all the microencapsulated extracts obtained, since the span values are close to 1 [63] and show that the type of encapsulating agent added to the extracts did not significantly affect the particle size distribution of the microparticles.

### 4.3. Thermal Behavior of Encapsulated Extract

The study of the thermal behavior of the extract is important as it provides information on the stability of the sample in the face of temperature variations, providing useful data for the quality control process. The first significant weight loss (50–120 °C) observed in the thermogravimetric curves corresponds to the water and volatile compounds loss by the samples, and the values of residual moisture content (2.90–4.25%) associated with this event were considered typical for microparticles obtained by spray-drying [64]. Similar results were observed in the microencapsulation of *Litsea glaucescens* infusions using maltodextrin DE 10 (1.82–4.32%) and for an encapsulated extract of *Hibiscus sabdariffa* using carboxymethylcellulose as the encapsulating agent (4.53%) [13,26]. This range of values has also been reported for microencapsulated extracts obtained from soybean using β-CD, maltodextrin, or Arabic gum as wall material (2.07–4.63%) and for the microencapsulation of bioactive compounds from *Camellia sinensis* using different biopolymers as encapsulating agents, including pectin (less than 2.40%) [65,66].

Despite the increase in the residual water content of the microencapsulated extracts (3.43–4.25%) compared to Bu-A (2.90%), all the samples had low levels of residual moisture, which tends to increase shelf life and reduce the possibility of microbiological contamination of the powders [55]. The statistical difference observed for Bu-MD11, Bu-Pec, and Bu-CMC compared to Bu-A (*p* < 0.05) can be attributed to the properties of the encapsulant agents, such as chemical structure and hygroscopicity. For example, MD11, composed of a mixture of glucose, disaccharides, and polysaccharides, has many hydroxyl groups capable of interacting with water, making this sugar more hygroscopic than MD4. According to Negrão-Murakami et al., this difference is noted because the maltodextrins with higher DE values, having more hydroxylated saccharides, show a chemical structure favorable for interaction with water [67]. In the case of pectin, this interaction is possibly due to the presence of hydroxyl, carboxyl, and amide groups (amidated pectin), which also give this excipient a certain degree of hygroscopicity [68]. Similarly, CMC is considered to be a highly hygroscopic excipient and can adsorb a large amount of water (>50%) under high-relative-humidity conditions [69]. According to Tonon et al., microparticles with many hydrophilic groups can easily bind water molecules from the ambient environment during sample handling after spray-drying, which could justify the higher residual moisture in the extracts prepared with these wall materials [48]. These authors have recorded similar results for *Euterpe oleraceae* microparticles obtained by atomization with maltodextrin DE 20 and Arabic gum.

The second event recorded in the TGA/DTG curves at 150 °C (Bu-A, EBu-MD4, Bu-MD11, and Bu-βCD) or at 190 °C (Bu-CMC and Bu-Pec) corresponds to the decomposition process of the samples. These results indicate that the microparticles obtained with Pec or CMC have a higher thermogravimetric stability than the others since they require a higher temperature to initiate the thermal decomposition step. These results are different from those observed by Nunes et al., who described an increase in the thermal stability of a concentrated extract of *Ilex paraguariensis* when encapsulated with different concentrations of maltodextrin [70]. However, it is noteworthy that the excipient concentrations chosen by the authors (20, 30, and 40%) were higher than the one used in the present work, which was only 10%. The loss of mass of the materials in this temperature range can be understood as a degradation due to the thermal decomposition of organic compounds and subsequent volatilization, a behavior also observed in the analysis of the by-product of cupuassu seeds and their respective crude extract when analyzed by using TGA/DTG [64].

Regarding the thermograms obtained by DSC, the first broad endothermic event (50–120 °C) corresponds to the moisture evaporation associated with the first part of the event of a glass transition in the microparticles, characteristic of amorphous materials, as is the case of dried plant extracts [30,71,72]. The statistical similarity between these values (*p* > 0.05) suggests that the encapsulating agents used did not modify this property of the microparticles. However, it should be considered that T_g_ is influenced by the amount of water present in the material analyzed, and this occurs at lower temperatures as the moisture content increases [73]. In the case of the microencapsulated extracts, which had different levels of residual water and were hygroscopic, the moisture adsorbed after spray-drying or during the preparation of the samples for analysis may have made it impossible to determine their glass transition temperatures accurately, so that the values are statistically similar, which would justify the observed results. The determination of T_g_ is important because the values of this parameter are directly related to the stability of the microparticle during storage. According to Gallo et al., structural changes occur in amorphous microparticles when they are stored at temperatures above the T_g_ (the temperature at which these materials transform from a glassy to a rubbery state) [74]. Therefore, microparticles with low moisture contents and T_g_ values above the storage temperature can be considered stable. For the samples analyzed, the glass transition temperature was always above room temperature, and so all microparticles showed adequate physical stability.

The second event in the thermograms may correspond to the onset of sample decomposition. The temperatures recorded for Bu-A, Bu-MD4, Bu-MD11, and Bu-βCD (around 160 °C) indicate that the addition of the encapsulating agents MD4, MD11, and βCD did not modify the thermal stability of Bu-L compared to the microparticles obtained without excipients (Bu-A). In contrast, the temperatures observed for Bu-Pec and Bu-CMC (191 °C and 188 °C, respectively) suggest that the microencapsulated extracts obtained with Pec and CMC were thermally more stable since their degradation process started at higher temperatures. These data are related to those obtained by using TG/DTG, where the changes in the baseline of the DSC curves were accompanied by a mass loss in the same temperature ranges, which reinforces the assumption that the second event corresponds to the degradation of samples.

### 4.4. Phenolic Compounds

Among the metabolites present in the samples, phenolic compounds were quantified at 69.98–101.28 mg GAEq/g Extract. These values are lower than those determined for extracts obtained from congeneric species, such as *B. vahlii* (237 mg GAEq/g) and *B. pulchella* (201.89 mg GAEq/g) [75,76]. On the other hand, the levels were higher than those determined for extracts of *B. forficata* (5.81–58.58 mg GAEq/g Extract), *B. scandens* (47.33 mg GAEq/g Extract), and *B. variegata* (69.39 mg GAEq/g Extract) [77,78,79]. These differences may be attributed not only to the different plant species but also to the type of solvents and extraction methods used, which would favor the obtaining of samples with different levels of phenolic compounds [78,80].

Regarding the total flavonoid content, the determined values (12.69–23.07 mg RUTE/g Extract) were similar to or higher than those recorded for aqueous extracts prepared with the leaves of *B. variegata* (3.86–18.40 mg RUTE/g Extract) [79]. In contrast, these values can be considered low compared to those determined for aqueous extracts prepared with the leaves of other species of the genus *Bauhinia*, such as *B. vahlii* (59 mg RUTE/g Extract) and *B. pulchella* (221.71 mg RUTE/g Extract) [75,76].

The similarity between the phenolic content of the lyophilized extract (Bu-L) and Bu-A (*p* > 0.05) allows us to affirm that the parameters used in the spray-drying process ensured the preservation of these compounds. The observed result agrees with that reported by Cunha et al., who obtained a spray-dried extract from the leaves of *B. forficata* with a preserved flavonoid profile under similar operating conditions to those used in the present work [33]. The spray-drying process can preserve the phenolic compounds present in an extract. Thus, these results confirm the applicability of the spray-drying technique for the atomization of thermosensitive samples, as is widely discussed in the literature [81,82,83]. The inlet temperature acts on the dryer evaporative capacity and thermal efficiency, while the outlet temperature controls the moisture content and surface morphology of the microparticles. The phenolic compounds were preserved even when using a high inlet temperature (150 °C) in the atomization of the extract solution. This was possible due to the rapid contact between the hot drying gas and the sample in the drying chamber after the atomization. Moreover, the outlet temperature, which is the temperature of the gas laden with solid particles before entering the cyclone, is much lower than the inlet temperature, preventing thermal degradation of the product.

Different behaviors were observed in the encapsulation of phenolic compounds by dextrin and the other encapsulating agents. Considering that Bu-L was added to the feed solutions to represent 90% of the solid content, this was the maximum YE value expected after the spray-drying process. The microparticles obtained with MD4, MD11, and βCD presented TPR values of 84.28, 90.99, and 84.78 mg GAEq/g Extract, corresponding to YE values of 83.09, 89.71, and 83.59%, respectively. These values are statistically similar to each other (*p* > 0.05), indicating the preservation of most of the phenolic compounds present in the atomized solution. Even in the case of TFC, the values obtained for Bu-MD4, BU-MD11, and Bu-βCD were the lowest among those recorded at around 12 mg RUTE/g Extract, with no statistically significant difference between them (*p* > 0.05). The reduction in TFC content may have been due to the loss of part of the flavonoid compounds during the spray-drying process or even during the extraction process used for the quantification of the phenolics in the microparticles.

However, in light of the results described above, it is possible to hypothesize that an interaction between dextrin and phenolic compounds in the SD microparticles occurred, mainly due to the non-flavonoid components of the phenols present. As reviewed by Jakobek and Matić, dextrin can bind to phenolic compounds, but there are still unknowns about this process [84]. For example, hydrogen bonds formed between the oxygen atoms of glycosidic bonds, present in dextrin, and the hydrogens of hydroxyl groups of phenolic compounds are possible interactions between maltodextrins and polyphenols. However, it should be considered that these interactions may vary depending on the affinity between the phenolic compounds and the encapsulating agent, as well as on properties such as the water solubility, molecular size, conformational mobility, and shape of the polyphenol. All these factors could justify the observed difference in the interaction between dextrin and flavonoid and non-flavonoid phenolic compounds present in the samples [85].

The TPR and TFC values recorded for Bu-Pec and Bu-CMC were exactly the opposite of those achieved with the maltodextrins: lower levels of phenolic compounds (69.98 and 75.15 mg GAEq/g Extract, respectively), resulting in a reduction in YE compared to the other microencapsulated extracts, a difference considered significant (*p* < 0.05), and higher TFC values among those obtained for the microencapsulated extracts, with amounts comparable or even higher than those obtained for Bu-L and Bu-A. In this case, a greater interaction of the wall materials with the flavonoids is suggested, which may have resulted in the loss of part of the non-flavonoid phenolic compounds during the spraying or extraction of substances from the particles for quantification, since the TPR was reduced and the TFC was maintained in these extracts after drying.

As for pectin and CMC, which are both considered dietary fibers, their interaction with polyphenols can occur through hydrogen bonding, van der Waals forces, and hydrophobic interactions. It can be said that polyphenols and fibers share some important general characteristics in these interactions. For fibers, OH and other functional groups are important, along with the degree of saturation, molecular weight, degree of aggregation, and structural and conformational organization. In the case of phenolic compounds, the presence of OH, CH_3,_ and galloyl groups; sugar molecules in the aglycone; flexibility and number of phenolic rings; molecular size; and spatial configuration have been suggested to influence dietary polyphenol–fiber interactions. In addition, their hydrophilic or hydrophobic character is important. Regarding the molecular size and number of phenolic rings present in polyphenols, which favor their interaction with Pec and CMC, flavonoids have a greater possibility of interaction compared to phenolic acids due to their structural characteristics, and this could justify the found results [84].

Despite the differences observed between the microparticle samples, the TPR values recorded for all the microencapsulated extracts were higher than those found by Port’s et al., which, when analyzing an infusion obtained from the leaves of the same variety of *B. ungulata*, quantified the phenolics in the sample at 23.67 mg AG/g [6]. However, it should be considered that the infusion prepared by these authors (500 mg in 25 mL of boiling water) produced a less concentrated aqueous extract than the one used in this work (1.25 g in 25 mL of boiling water), so a lower content of total phenolic compounds could be expected.

In quantitative terms, only the contents of chlorogenic acid and rutin were variable. The extract sprayed without wall materials (Bu-A) showed values significantly higher than those determined for the microparticles (*p* < 0.05), but this result was expected since the addition of encapsulating agents reduces the content of the compounds present in the microparticles, which are composed of 90% (*w*/*w*) Bu-L and 10% (*w*/*w*) wall material. In the case of Bu-A, the values determined were expected to be equivalent to those obtained for the extract, since the microparticle was obtained from the lyophilized extract. However, regarding the fact that plant extracts are heterogeneous samples, even considering the sample moisture, the dry extract weighed to prepare the spraying solution could have contained residual moisture values higher than those assessed (7.9%), which could explain the lower content of chlorogenic acid and rutin in the lyophilized extract. Among the microparticles, Bu-Pec and Bu-CMC were the samples with the highest content of chlorogenic acid and rutin, as both were significantly different from all the other samples (*p* < 0.05), except for Bu-βCD, which showed comparable levels of these compounds.

The chlorogenic acid content determined in the samples (1.740 to 1.975 mg/g Extract) is close to that observed in a polyphenolic fraction obtained from *Catharanthus roseus* stems (2.16 mg/g). This sample was effective in reducing the glycemia of normal or diabetic mice 6 h after its administration, in addition to stimulating insulin secretion in RINm5F cells [86]. Similarly, an aqueous extract of *Cecropia obtusifolia* leaves standardized in chlorogenic acid (5.2 µg/g) promoted, among other effects, an increase in insulin secretion in vitro and acute and subacute reductions in glucose levels in diabetic mice. In addition, daily administration of *C. obtusifolia* increased hepatic glycogen storage and glycogen synthase levels in animals without causing apparent changes in gluconeogenesis [87]. It is worth noting that the chlorogenic acid content determined by the authors was much lower than that found for the aqueous extract of *Bauhinia ungulata* var. *obtusifolia*.

As for p-coumaric acid, the levels determined in the aqueous extract and SD microparticles (0.060–0.078 mg/g extract) were higher than the value determined for an aqueous extract obtained from *Cucurbita ficifolia* fruits, which contained this phenolic as one of its five main compounds (58 µg/g). This sample showed an acute and sub-chronic hypoglycemic effect when administered to normal or diabetic mice, also promoting a greater accumulation of glycogen in the liver of the animals, increased levels of glycogen synthase, and decreased glycogen phosphorylase enzyme [88].

The values of rutin measured in the samples (11.216–12.886 mg/g Extract) were remarkably close to those found by Gandhi et al., which quantified the flavonoid at 1.36% (*w*/*w*) (equivalent to 13.6 mg/g) in the methanolic extract prepared from the fruit of *Solanum torvum* Swartz. Diabetic rats were characterized as hypoglycemic after its administration by induction with streptozotocin [89]. These values were even higher than those determined in extracts obtained from the leaves of *Cnidoscolus chayamansa* (2.00 mg/g) and *Bauhinia variegata* (4.38 mg/g), both of which were able to reduce fasting blood glucose when administered to diabetic rats. [90,91]

Regarding isoquercitrin, the contents (0.485–0.533 mg/g Extract) were in the range of values determined in derivatives obtained from *Ribes meyeri* leaves (0.09–14.64 mg/g). These samples were effective in increasing glucose uptake in 3T3-L1 adipocytes, suggesting their potential use as a functional food ingredient for the prevention of type 2 diabetes [92].

### 4.5. Encapsulation Efficiency

Bu-MD11, Bu-CMC, and Bu-βCD showed better phenol encapsulation efficiency, indicating that a greater amount of such substances is located inside the particle in these extracts, which favors their preservation and tends to increase the stability of the microparticles [56,93].

Unexpectedly, the microparticles obtained with MD11 showed a higher encapsulation efficiency than that observed for Bu-MD4. The fact that maltodextrin DE 11-14 presents a higher amount of low-molecular-weight sugars compared to MD4 suggests that its ability to retain phenolic compounds inside the particles is lower, which would reduce its encapsulation efficiency, as already reported by other authors [67,94]. However, it was observed that this polymer had a lower PS/TPR ratio, resulting in a higher encapsulation efficiency of the phenolic compounds of *B. ungulata*. This phenomenon may be due to the agglomeration of the particles resulting from the use of MD11, as observed in the SEM images and from the particle size distribution analysis. This agglomeration hinders the extraction of compounds present on the surface since they tend to form larger structures with less contact surface, where phenolic compounds tend to be trapped. Similar results were found by Etzbach et al. in the encapsulation of *Physalis peruviana* L. by spray-drying using different encapsulating agents [93]. The authors observed that the particle size had a strong influence on the encapsulation efficiency, as the fine microparticles, due to their greater surface area, showed a significantly higher content of the compounds on the surface and lower encapsulation efficiency compared to the agglomerated microparticles.

Encapsulation efficiency is an important indicator in particle analysis and refers to the potential of wall materials to encapsulate or maintain the material in the core of the formed structure [95]. In terms of this parameter, Bu-MD11, Bu-CMC, and Bu-βCD proved to be the most advantageous, also showing high encapsulation yields of phenolic compounds. Among these samples, Bu-CMC was the microencapsulated extract that, in addition to the advantages already described, presented the highest content of total flavonoids, presenting itself as an interesting alternative in the microencapsulation of aqueous extracts of *B. ungulata* var. *obtusifolia*.

### 4.6. Antioxidant Activity

The samples scavenged DPPH^∙^ and ABTS^+^ radicals with different scavenging percentages and TAA. This difference was expected due to the different affinities of the radicals used by the substances present in the extracts. While ABTS^+^ radicals have an affinity for hydrophilic, lipophilic, and hydrogen atom donor compounds, DPPH^∙^ interacts more with lipophilic compounds, showing a lower affinity for compounds containing aromatic rings with only hydroxyl groups [96,97]. In this sense, TAA tends to be higher with the ABTS^∙+^ radical scavenging method and lower when the DPPH^∙^ radical is used, in agreement with what has been observed in the present work. Similar results were found by Floegel et al., who compared the antioxidant activity of the 50 most popular antioxidant-rich fruits, vegetables, and beverages in the US diet using the ABTS and DPPH assays. They reported that the ABTS assay better reflects the antioxidant activity of various foods than the DPPH assay [98].

Regarding DPPH^∙^ scavenging, Bu-L had the lowest total antioxidant activity compared to the others (308.67 ± 2.64 µm TEq/g Extract), but this value was not significantly different from Bu-A (*p* > 0.05), indicating the preservation of this property in the atomized extract without wall materials. All microencapsulated extracts had higher TAA values than the unencapsulated samples, indicating that the addition of encapsulating agents increased the ability of the extract to scavenge DPPH^∙^ radicals. This result may be justified by the fact that the wall materials used in this study are described in the literature as antioxidants or secondary antioxidants, each capable of interacting with different types of radicals at different intensities and through different mechanisms [99,100,101,102]. In this sense, these agents, together with the compounds present in Bu-L, may have contributed to the scavenging of DPPH^∙^ radicals.

Bu-MD11 and Bu-βCD were the most active samples, and the TAA values were not statistically different between them. The TAA value for Bu-CMC was lower than those measured for Bu-MD11 and Bu-βCD, but not statistically different. The lyophilized extract (Bu-L), the SD microparticles without the wall material (Bu-A), and Bu-MD4 were the less active samples and were statistically different from Bu-MD11, Bu-βCD, and Bu-CMC (*p* > 0.05). Bu-Pec was placed in an intermediate position between the three most active samples and the three least active samples, although it was not statistically different from Bu-CMC.

The best results obtained for the spray-dried samples prepared with MD11, βCD, and CMC may be related to the higher encapsulation efficiency determined for these extracts, which, due to this property, were able to preserve the phenolic compounds proposed to be involved in the scavenging of DPPH^∙^ radicals inside the particles [56,95].

In terms of ABTS^+^ radical scavenging, Bu-CMC and Bu-Pec were the most active, showing that the use of CMC and Pec increased the TAA of these samples compared to the others. Contrary to what was expected, these microparticles presented the lowest TPR, an unusual result since phenolic compounds have been positively correlated with the antioxidant capacity of plant extracts by different analytical methods [103,104]. However, it is important to note that this correlation is not always observed. For example, in the work of Rivero-Pérez et al., which consisted of the analysis of 321 wine samples, many of those with a high content of phenolic compounds showed low values of total antioxidant capacity [105]. It was suggested that this result could be justified by the fact that the antioxidant capacity of wine is more related to the type of phenolic compounds present in wines than to their total content. According to the authors, this property is due to the flavonoids present in the beverage, especially tannins and anthocyanins, and the tested samples contain low levels of these compounds due to their method of preparation, which affects their antioxidant activity.

Similarly, Bu-CMC and Bu-Pec, despite having the lowest TPR values, contain high TFC, which leads to the understanding that their higher AAT values may have mainly been related to the flavonoids present in these samples. As reviewed by Rice-Evans et al., the flavonoids are electron/H^+^ donors due to the reduction of several hydroxyl groups present in their structure, which is related directly to the antioxidant activity of these phenolics [106]. In addition, among the structural characteristics that contribute to the action of flavonoids as antioxidants are the presence of catechol or dihydroxy structures in the B ring; the presence of a double bond in the C ring between carbons C2-C3, in conjugation with the ketone function in C4; and the presence of hydroxyl groups in C3 and C5. These factors allow for the stability of the molecule since they favor an extensive delocalization of electrons in the aromatic nuclei, different from that observed in other phenolic compounds, such as phenolic acids derived from hydroxybenzoic or hydroxycinnamic acids. These structural characteristics of the flavonoids could explain the observed result.

The microparticles obtained with CMC and Pec showed better antioxidant activity, even with low TPR. These data have allowed us to conclude that the phenolic compounds present in lower amounts in these samples are non-flavonoids. Based on these results, it is possible to affirm that among the microencapsulated extracts, Bu-CMC and Bu-Pec proved to be worthwhile antioxidants. This characterization reinforces the antioxidant potential of the genus and suggests that this variety of *B. ungulata* can be employed as a natural antioxidant. As a plant popularly used as a hypoglycemic agent, this activity becomes even more important, since oxidative stress resulting from the generation of free radicals in diabetes is related to the development of secondary complications in patients with the disease, arousing interest in new therapeutic options that, in addition to acting as hypoglycemic agents, can protect the body from the action of these radicals [107].

## 5. Conclusions

In this work, microparticles were prepared from an aqueous extract of *Bauhinia ungulata* var. *obtusifolia* leaves by spray-drying and associated with maltodextrin DE 11-14, and 4-7, β-cyclodextrin, pectin, and sodium carboxymethylcellulose to protect their phenolic compounds, which exhibit antioxidant and antidiabetic activities, among them p-coumaric acid, chlorogenic acid, rutin, and isoquercitrin, whose contents are comparable to or higher than those described for other antidiabetic species. This fact allows for the inference that the reported species shows viability for use to treat diabetes, as stated in the literature about its folk use. The comparison of total phenolic and total flavonoid contents determined before and after atomization and the quantification of each of the already mentioned compounds in these samples indicates that the phenolic compounds were preserved during the operation, indicating the stability of the extract. As expected, the samples demonstrated antioxidant capacity due to the presence of the flavonoids that were detected and quantified, and this finding allows us to state that this variety of *B. ungulata*, used as a natural antioxidant, has increased efficacy in the treatment of diabetes since oxidative damage is involved in the complications of this disease. CMC showed high suitability for microencapsulation of the extract, yielding good spray-drying yield, adequate particle morphology and size, increased thermal stability, good EE, high TFC, and the highest antioxidant activity and content of substances. However, particle agglomeration, low TPR, and a consequent reduction in the encapsulation yield also occur through the use of this wall agent. These problems could be overcome by combining CMC and MD4, since Bu-MD4 showed a high spray-drying yield and monomodal particle distribution, indicating less particle agglomeration, and was among the samples with the highest TPR, thus presenting a high YE. The results reported in the present work provide relevant data on *Bauhinia ungulata* var. *obtusifolia* (Ducke) Vaz, a plant species on which, to date, no knowledge about the investigation of its leaves’ aqueous extracts under the various aspects analyzed has been published in the literature, including microencapsulation and chemical constitution.

## Figures and Tables

**Figure 1 pharmaceutics-16-00488-f001:**
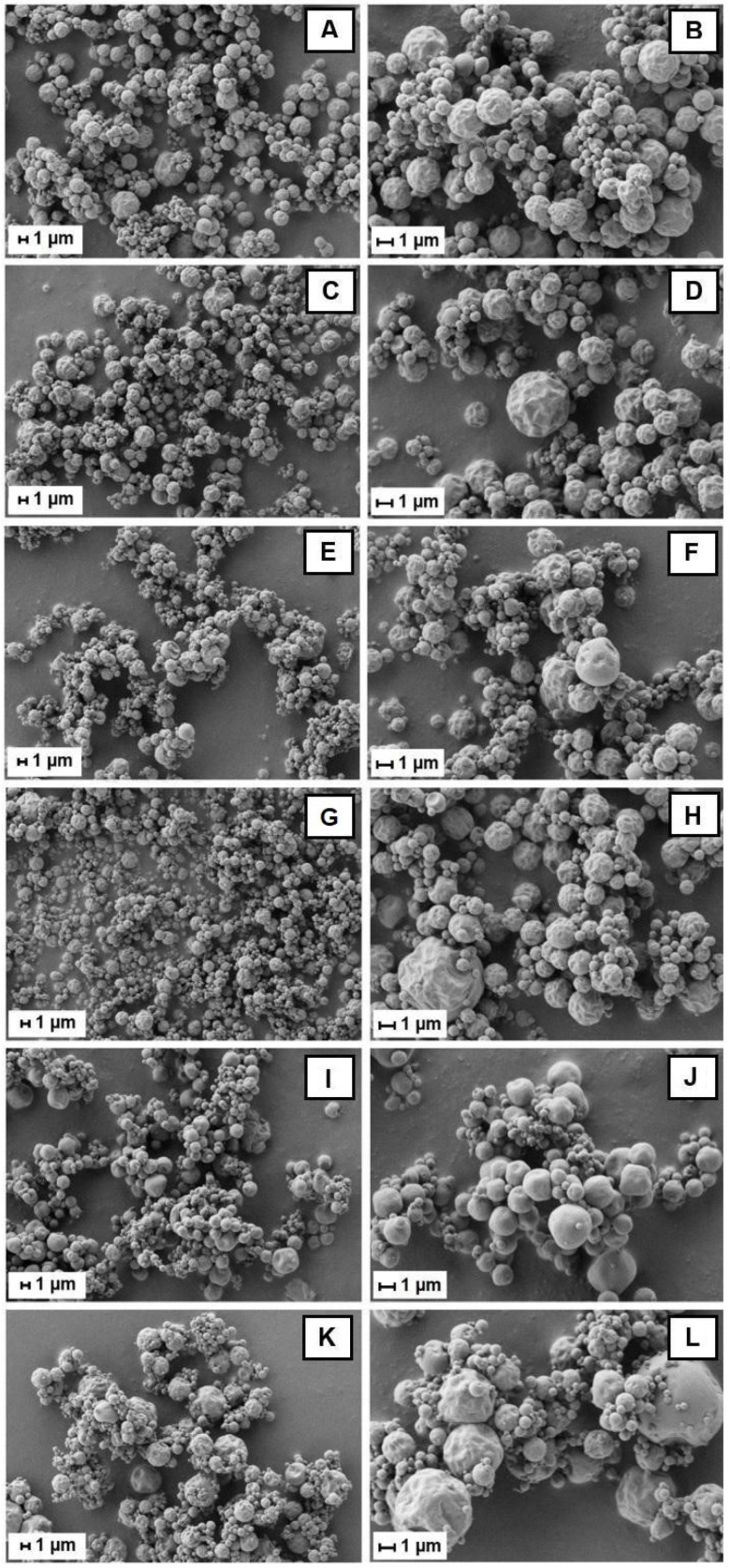
SEM images of Bu-A (**A**,**B**), Bu-MD4 (**C**,**D**), Bu-MD11 (**E**,**F**), Bu-βCD (**G**,**H**), Bu-Pec (**I**,**J**), and Bu-CMC (**K**,**L**) at magnifications of 5000X (**left**) and 10,000X (**right**).

**Figure 2 pharmaceutics-16-00488-f002:**
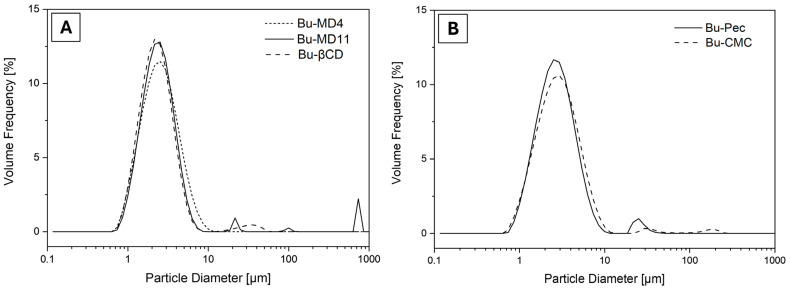
Particle size distribution of (**A**) Bu-MD4, Bu-MD11, Bu-βCD, and (**B**) Bu-Pec and Bu-CMC.

**Figure 3 pharmaceutics-16-00488-f003:**
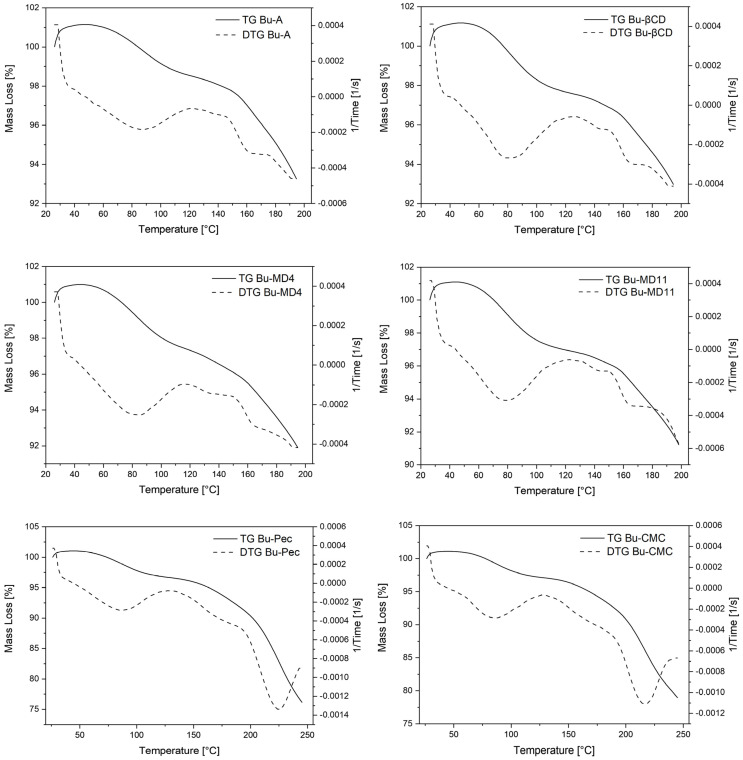
Thermograms by TGA/DTG of Bu-A, Bu-βCD, Bu-MD4, Bu-MD11, Bu-Pec and Bu-CMC.

**Figure 4 pharmaceutics-16-00488-f004:**
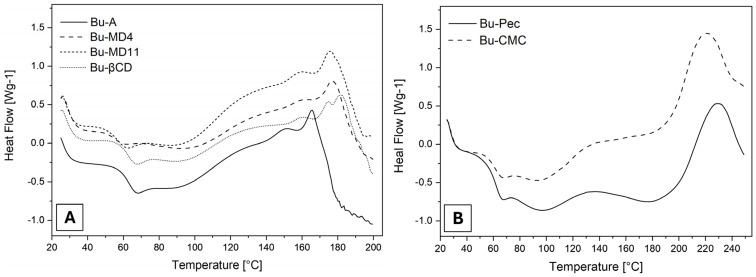
DSC thermograms of (**A**) Bu-A, Bu-MD4, Bu-MD11, Bu-βCD, and (**B**) Bu-Pec and Bu-CMC (^exo).

**Table 1 pharmaceutics-16-00488-t001:** The concentration of standard compounds in the working solutions.

Solutions	Concentration (µg/mL)			
	Chlorogenic Acid	p-Coumaric Acid	Rutin	Isoquercitrin
1	50	5.00	200	15.00
2	25	2.50	100	7.50
3	10	1.00	40	3.00
4	5	0.50	20	1.50
5	2.5	0.25	10	0.75

**Table 2 pharmaceutics-16-00488-t002:** Solid content and viscosity (mean values ± standard deviation, n = 3) of the solutions to be sprayed without (Bu-A) or with wall material (Bu-MD4, Bu-MD11, Bu-βCD, Bu-Pec, and Bu-CMC). The yield of the spray-drying process (YD, %) is indicated by the values obtained from two replicates of the experiments. The lowercase letters refer to the statistical analysis (see Section 2.2.12).

Solution	Wall Material	Solid Content (%)	Viscosity (mPa.s)	YD (%)
Bu-A	None	0.8	1.45 ± 0.03 ^a^	64.71; 63.69
Bu-MD4	Maltodextrin DE 4-7	0.9	1.44 ± 0.04 ^a^	89.03; 88.01
Bu-MD11	Maltodextrin DE 11-14	0.9	1.43 ± 0.03 ^a^	74.13; 80.68
Bu-βCD	β-cyclodextrin	0.9	1.43 ± 0.02 ^a^	73.45; 83.95
Bu-Pec	Pectin LM-22-CG	0.9	1.41 ± 0.01 ^a^	72.47; 73.90
Bu-CMC	Sodium carboxymethylcellulose	0.9	3.46 ± 0.03 ^b^	77.40; 75.21

**Table 3 pharmaceutics-16-00488-t003:** Particle size distribution of the SD microparticles containing the wall material (mean ± standard deviation, n = 3). The lowercase letters refer to the statistical analysis (see Section 2.2.12).

Sample	Dv_10_ (µm)	Dv_50_ (µm)	Dv_90_ (µm)	Span
Bu-MD4	1.22 ± 0.02 ^a^	2.23 ± 0.17 ^ab^	4.66 ± 0.71 ^ab^	1.53 ± 0.21 ^a^
Bu-MD11	1.19 ± 0.08 ^a^	2.41 ± 0.39 ^ab^	4.54 ± 0.79 ^ab^	1.40 ± 0.30 ^a^
Bu-βCD	1.15 ± 0.07 ^a^	2.01 ± 0.13 ^a^	3.76 ± 0.40 ^a^	1.29 ± 0.10 ^a^
Bu-Pec	1.29 ± 0.02 ^a^	2.50 ± 0.02 ^ab^	4.82 ± 0.11 ^ab^	1.41 ± 0.04 ^a^
Bu-CMC	1.29 ± 0.07 ^a^	2.87 ± 0.25 ^b^	5.54 ± 0.34 ^b^	1.50 ± 0.23 ^a^

**Table 4 pharmaceutics-16-00488-t004:** Moisture content, glass transition, and initial decomposition temperature of the spray-dried extracts (mean values ± standard deviation, n = 3). The lowercase letters refer to the statistical analysis (see Section 2.2.12).

Sample	Moisture Content (%)	T_g_ (°C)	T_initial_ Decomposition (°C)
Bu-A	2.90 ± 0.10 ^a^	60.4 ± 3.5 ^a^	160.2 ± 1.6 ^a^
Bu-MD4	3.43 ± 0.16 ^ab^	56.8 ± 2.9 ^a^	165.8 ± 3.6 ^a^
Bu-MD11	3.89 ± 0.27 ^b^	57.6 ± 2.4 ^a^	163.6 ± 3.9 ^a^
Bu-βCD	3.63 ± 0.27 ^ab^	61.5 ± 0.4 ^a^	162.6 ± 6.4 ^a^
Bu-Pec	4.05 ± 0.40 ^b^	58.4 ± 4.8 ^a^	191.4 ± 0.9 ^b^
Bu-CMC	4.25 ± 0.48 ^b^	57.8 ± 5.6 ^a^	188.2 ± 1.1 ^b^

**Table 5 pharmaceutics-16-00488-t005:** Total flavonoid content (TFC), total phenolic content retained in the microparticles (TPR), total phenolic content on the surface of the microparticles (PS), encapsulation efficiency (EE), and encapsulation yield (YE) of the extracts (mean values ± standard deviation, n = 3). The lowercase letters refer to the statistical analysis (see Section 2.2.12).

Sample	TFC (mg RUTE/g Extract)	TPR (mg GAEq/g Extract)	PS (mg GAEq/g Extract)	EE (%)	YE (%)
Bu-L	21.93 ± 0.30 ^a^	101.43 ± 1.68 ^a^	⎯	⎯	⎯
Bu-A	22.69 ± 0.45 ^ac^	101.28 ± 1.65 ^a^	⎯	⎯	⎯
Bu-MD4	12.75 ± 0.57 ^b^	84.28 ± 5.47 ^b^	38.69 ± 0.64 ^a^	44.95 ± 5.17 ^a^	83.09 ± 5.40 ^a^
Bu-MD11	12.69 ± 0.21 ^b^	90.99 ± 2.31 ^b^	32.82 ± 2.51 ^ab^	57.35 ± 2.24 ^b^	89.71 ± 2.28 ^a^
Bu-βCD	12.69 ± 0.31 ^b^	84.78 ± 1.70 ^b^	33.57 ± 1.28 ^ab^	50.49 ± 2.92 ^ab^	83.59 ± 1.68 ^a^
Bu-Pec	21.73 ± 0.34 ^a^	69.98 ± 3.44 ^c^	33.59 ± 3.87 ^ab^	35.87 ± 3.49 ^c^	68.99 ± 3.39 ^b^
Bu-CMC	23.07 ± 0.50 ^c^	75.15 ± 3.13 ^c^	27.96 ± 1.99 ^b^	46.53 ± 1.13 ^a^	74.09 ± 3.09 ^b^

**Table 6 pharmaceutics-16-00488-t006:** Scavenging percentages of DPPH^∙^ and ABTS^∙+^ radicals by extracts and TAA (mean values ± standard deviation, n = 3). The lowercase letters refer to the statistical analysis (see Section 2.2.12).

Sample	DPPH^∙^		ABTS^∙+^	
	Scavenging Percentage (%)	TAA (µM TEq/g Extract)	Scavenging Percentage (%)	TAA (µM TEq/g Extract)
Bu-L	22.09 ± 0.66 ^a^	308.67 ± 2.64 ^a^	23.97 ± 0.22 ^a^	956.40 ± 2.81 ^a^
Bu-A	25.65 ± 0.25 ^b^	323.45 ± 1.07 ^a^	29.49 ± 0.55 ^b^	1031.33 ± 8.04 ^b^
Bu-MD4	30.41 ± 0.60 ^c^	345.58 ± 2.98 ^b^	23.84 ± 1.89 ^a^	955.17 ± 23.65 ^a^
Bu-MD11	38.34 ± 1.04 ^d^	390.06 ± 6.66 ^c^	25.24 ± 1.34 ^a^	972.82 ± 17.26 ^a^
Bu-βCD	38.22 ± 0.43 ^d^	389.26 ± 2.72 ^c^	24.58 ± 1.26 ^a^	964.31 ± 16.12 ^a^
Bu-Pec	34.61 ± 1.17 ^e^	367.86 ± 6.65 ^d^	31.54 ± 0.77 ^bc^	1062.32 ± 11.94 ^bc^
Bu-CMC	36.10 ± 1.74 ^de^	376.55 ± 10.16 ^cd^	33.02 ± 0.92 ^c^	1085.72 ± 14.84 ^c^

**Table 7 pharmaceutics-16-00488-t007:** Content (µg/mL or mg/g Extract) of chlorogenic acid, p-coumaric acid, rutin, and isoquercitrin in the extract (mean values ± standard deviation, n = 3). The lowercase letters refer to the statistical analysis (see Section 2.2.12).

Sample	Content	Compound			
		Chlorogenic Acid	p-Coumaric Acid	Rutin	Isoquercitrin
Bu-L	µg/mL	15.38 ± 0.22 ^a^	0.60 ± 0.05 ^a^	102.02 ± 2.52 ^a^	4.30 ± 0.21 ^a^
Bu-A		19.17 ± 0.51 ^b^	0.76 ± 0.08 ^a^	125.13 ± 1.12 ^b^	5.17 ± 0.67 ^a^
Bu-MD4		17.08 ± 0.06 ^cd^	0.58 ± 0.11 ^a^	108. 30 ± 0.60 ^c^	4.82 ± 0.27 ^a^
Bu-MD11		16.72 ± 0.46 ^d^	0.69 ± 0.01 ^a^	110.19 ± 1.58 ^cd^	4.66 ± 0.40 ^a^
Bu-βCD		17.43 ± 0.15 ^cde^	0.60 ± 0.04 ^a^	113.07 ± 0.94 ^de^	4.90 ± 0.42 ^a^
Bu-Pec		17.90 ± 0.12 ^e^	0.70 ± 0.04 ^a^	115.10 ± 0.76 ^e^	5.01 ± 0.12 ^a^
Bu-CMC		17.80 ± 0.15 ^ce^	0.68 ± 0.10 ^a^	112.59 ± 1.47 ^de^	4.84 ± 0.30 ^a^
Bu-L	mg/g Extract	1.669 ± 0.024 ^a^	0.066 ± 0.005 ^a^	11.077 ± 0.274 ^a^	0.466 ± 0.022 ^a^
Bu-A		1.975 ± 0.052 ^b^	0.078 ± 0.008 ^a^	12.886 ± 0.115 ^b^	0.533 ± 0.069 ^a^
Bu-MD4		1.769 ± 0.007 ^c^	0.060 ± 0.012 ^a^	11.216 ± 0.062 ^a^	0.500 ± 0.029 ^a^
Bu-MD11		1.740 ± 0.048 ^ac^	0.072 ± 0.001 ^a^	11.465 ± 0.164 ^ac^	0.485 ± 0.041 ^a^
Bu-βCD		1.808 ± 0.016 ^cd^	0.062 ± 0.004 ^a^	11.724 ± 0.098 ^cd^	0.508 ± 0.044 ^a^
Bu-Pec		1.867 ± 0.013 ^d^	0.073 ± 0.004 ^a^	12.003 ± 0.080 ^d^	0.522 ± 0.013 ^a^
Bu-CMC		1.858 ± 0.016 ^d^	0.070 ± 0.011 ^a^	11.755 ± 0.153 ^cd^	0.505 ± 0.032 ^a^

## Data Availability

The data presented in this study are available in this article (and Appendix A).

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
