# Peer review of "Spray-Drying Microencapsulation of Bauhinia ungulata L. var. obtusifolia Aqueous Extract Containing Phenolic Compounds: A Comparative Study Using Different Wall Materials"

_pharmaceutics, 2024, doi:10.3390/pharmaceutics16040488_

Round 1

Reviewer 1 Report

Comments and Suggestions for Authors

1. Some references need to be updated.

2. The figures are not clear, please revise.

Comments on the Quality of English Language

1. The grammar of the whole manuscript needs to be revised carefully.

Author Response

The authors are grateful and thank the Reviewer for the feedback and the especially useful comments for the revision of the manuscript.

Reviewer 2 Report

Comments and Suggestions for Authors

A solid wor. I have few remarks and requests only:

1. Through the paper Authors use as synonyms microcapsules and microparticles. Based on the available characterization one cannot distinguish between microspheres (more likely here) and microcapsules - I'd sugggest to stay with microparticles

2. I couldn't find a clear statement which material is the best to co-spray with the extract - please provide it in coclusions and/or discussion - for the latter some justification should be presented

3. I strongly protest against mean and SD for n=2 (Table2, where is Table1?). This is simply impossible to calculate given 2 cases. Please get rid of this statement as well as any assesment of significance of yield

Author Response

(The authors gave the same response as above.)

Reviewer 3 Report

Comments and Suggestions for Authors

In the manuscript by Ramos Barbosa et al. entitled »Spray-drying microencapsulation of phenolic compounds from Bauhinia ungulata L. var. obtusifolia aqueous extract: a comparative study using different wall materials«, the authors investigate the suitability of different wall materials for the encapsulation of phenolic compounds from Bauhinia ungulata. The paper is written in readable English and the conclusions seem to be generally correct.

I recommend publication of the paper after some suggestions for minor improvements have been considered:

- The discussion section is quite long, and I suggest the authors to divide this part into a few more subsections

- At the beginning of the part »2.1 Properties of extracts and spray-drying solutions« it is stated: »The lyophilized extract obtained by freeze-drying the aqueous extract, Bu-L, presented 7.9% w/w of residual moisture. The solutions used for spray drying contained 51 mg of wall material, showed a solid content of approximately 9.1 g/L (Table 2), and viscosity for all samples quite similar, except for Bu-CMC, whose value was higher (p < 0.0001).« From this information, I can not figure out what the content of the wall material was. I know it says in »4.2. Preparation of the encapsulated samples« that the solutions contained 51 mg of the encapsulating agents, 56 mL of purified water, and 500 mg of the extract but such information should perhaps also be included in part 2.1. At this point I also have a question for the authors: How the volumes of the solutions for atomization were obtained with precision of four significant digits (to calculate the solids content to four digits)? This could be an exaggeration as the water content in the extract was not determined with such accuracy either.

- I know that NaCMC is often used in the pharmaceutical industry as an excipient in tablets. However, is it known whether NaCMC is harmless during digestion? After all, this material is a chemically modified natural polymer, which as such (after partial enzymatic degradation) is a foreign molecule in human metabolism.

Comments on the Quality of English Language

Author Response

(The authors gave the same response as above.)

Reviewer 4 Report

Comments and Suggestions for Authors

Generally, the authors present an interesting study. However, there are some problems in this paper and it may be considered for publication with some correction. Please find my specific comments below:

1.      The title, did the authors studied the spray-drying microencapsulation of ‘phenolic compounds’ or spray-drying microencapsulation of the whole extract which contained phenolic compounds and other compounds?

2.      In section 2.1, what was the solution used for spray-drying?

3.      In section 2.2, the authors mentioned ‘All samples had a YD greater than 60% (Table 2), with no statistical difference between microparticles and Bu-A (p>0.05)’. However, in Table 2, there was statistical difference between Bu-MD4 and Bu-A. Please double check all the results.

4.      Table 3, please indicate the meaning of different letters and double check the assignment of the letters was correct or not.

Author Response

(The authors gave the same response as above.)

Reviewer 5 Report

Comments and Suggestions for Authors

Authors developed spray-drying microparticles encapsulated with phenolic by using different wall materials. Even with many of funding, there are lots of issues need to be addressed before accepting.

1.     Lots of sentence are hard to understand. Please double check each sentence and paragraph. Moreover, this research manuscript discussion part contains 11 pages, which is too long, please short the discussion part.

2.     Table 3, please give more details, please explain what are Dv10, Dv90, “a”, “b”, “abc”…. Table 6 also.

3.     Authors mentioned that all samples have higher of PDI. It would be better to include the exact number of PDI in table 3.

4.     This manuscript contains lot of abbreviations, please give the full name when first using it.

5.     Please include all the formulars for calculation of TFC, TPC, YE. Please also give the methods for measuring the phenolic compounds on their surface (PS).

6.     What about the stability of the extract during the spray-drying.

7.     Did authors measure the particles size immediately after the spry-drying, before ultrasonication? Why cyclohexane was used for preparing the suspension, not PBS or water? Those microparticles should be water soluble.

Comments on the Quality of English Language

Extensive editing of English language required

Author Response

(The authors gave the same response as above.)

Round 2

Reviewer 2 Report

Comments and Suggestions for Authors

I wanted to thank Authors for complying with my remarks. It is a go. Congrats!

Author Response

The team of authors thanks you for your work which contributed immensely to increasing the quality of our paper, cordial greetings from the Amazon

Reviewer 4 Report

Comments and Suggestions for Authors The authors have tried to adjust the text to be clearer. Most questions raised by the reviewers are well answered. I believe the manuscript has been significantly improved and now warrants publication in Pharmaceutics.

Author Response

The team of authors thanks you for your work which contributed immensely to increasing the quality of our paper, cordial greetings from the Amazon!